# How intramyocardial fat can alter the electric field distribution during Pulsed Field Ablation (PFA): Qualitative findings from computer modeling

Juan J. Pérez[1], Ana González-Suárez[2,3]*

1 BioMIT, Department of Electronic Engineering, Universitat Politècnica de València, Valencia, Spain,
2 Translational Medical Device Lab, School of Engineering, University of Galway, Galway, Ireland,
3 Universidad Internacional de Valencia—VIU, Valencia, Spain

* ana.gonzalezsuarez@universityofgalway.ie

## Abstract

Even though the preliminary experimental data suggests that cardiac Pulsed Field Ablation (PFA) could be superior to radiofrequency ablation (RFA) in terms of being able to ablate the viable myocardium separated from the catheter by collagen and fat, as yet there is no formal physical-based analysis that describes the process by which fat can affect the electric field distribution. Our objective was thus to determine the electrical impact of intramyocardial fat during PFA by means of computer modeling. Computer models were built considering a PFA 3.5-mm blunt-tip catheter in contact with a 7-mm ventricular wall (with and without a scar) and a 2-mm epicardial fat layer. High voltage was set to obtain delivered currents of 19, 22 and 25 A. An electric field value of 1000 V/cm was considered as the lethal threshold. We found that the presence of fibrotic tissue in the scar seems to have a similar impact on the electric field distribution and lesion size to that of healthy myocardium only. However, intramyocardial fat considerably alters the electrical field distribution and the resulting lesion shape. The electric field tends to peak in zones with fat, even away from the ablation electrode, so that 'cold points' (i.e. low electric fields) appear around the fat at the current entry and exit points, while 'hot points' (high electric fields) occur in the lateral areas of the fat zones. The results show that intramyocardial fat can alter the electric field distribution and lesion size during PFA due to its much lower electrical conductivity than that of myocardium and fibrotic tissue.

**Data Availability Statement:** All relevant data are within the paper and its Supporting Information files.

## Introduction

Pulsed field ablation (PFA) is an interventional treatment method for arrhythmia and consists of applying short-duration high voltage pulses through ablation catheters. The PFA lesion is created by irreversible electroporation (IRE), which results in cell death [1]. As one of PFA's potential benefits over other thermal ablation techniques (such as radiofrequency ablation, RFA), different tissues have been claimed to present different sensitivities to IRE and the

**Funding:** Funded by the Spanish Ministerio de Ciencia e Innovación / Agencia Estatal de Investigación (MCIN/AEI/10.13039/501100011033) with Grant number PID2022-136273OB-C31 and PID2022-136273OA-C33. The funders had no role in study design, data collection and analysis, decision to publish, or preparation of the manuscript.

**Competing interests:** The authors have declared that no competing interests exist.

myocardium has been shown to be the most susceptible [2]. However, besides this tissue-specific sensitivity (which is exclusively dependent on the response of the different cell types to the electric field), the tissue's passive electrical properties and their relative spatial distribution will also affect the resulting distribution of the electric field. This issue is possibly a key factor when PFA is conducted on extremely heterogeneous substrates such as a healed infarction in the context of the catheter ablation of scar-mediated ventricular tachycardia (VT) and epicardial fat attached to the atrial wall in catheter ablation of atrial fibrillation (AF). The experimental data comparing PFA lesions on healthy myocardium and scar (healed infarction) tend to be rather scarce and are limited to pre-clinical models [3–5]. The preliminary results suggest that a superficial scar does not significantly impair PFA and creates lesion depths in healthy myocardium similar to adjacent healed endocardial scars (4.8 and 5.6 mm, respectively) [3]. Although the preliminary experimental data suggests that PFA may be superior to RFA in terms of being able to ablate viable myocardium separated from the catheter by collagen and fat [5], there is as yet no formal analysis based on physical laws that describes the process by which intramyocardial fat can affect the electric field distribution and alter the size of the PFA-induced lesion. Since fat is one order of magnitude less conductive than fibrotic and myocardial tissue (0.08 vs. 0.85 and 0.6 S/m, respectively), its presence could notably alter the electric field distribution and therefore the PFA-induced lesion size. Our objective was therefore to determine the electrical impact of the fat during PFA by means of computer modeling. This fat, for example, can be both intramyocardial fat deposited in a scar after a heart infarction or epicardial fat (i.e. between the heart and the pericardium).

## Methods

### Description of the computer model

Computer models were built considering a blunt-tip ablation catheter of 7-Fr 3.5-mm perpendicular to a ventricular endocardium, as proposed in Verma *et al* [6], inserted 0.5 mm into the tissue and surrounded by circulating blood, including a 2-mm epicardial fat layer and a 7 mm-thick cardiac wall (value based on that measured by CT angiography for the left ventricular mid-diastolic wall, 7.24 ±1.86 mm [7]). The rest of the model was filled with connective tissue composed of a mix of skeletal muscle and fat, as in Irastorza *et al* [8]. The ventricular wall comprised three types of tissue (viable myocardium, fibrous tissue and fat) following a spatial distribution based on a microscopic image reported in Sasaki *et al* [9] and used in a previous radiofrequency ablation computer modeling study [10]. For the sake of computational simplicity, the model volume was built by rotating this image over a symmetry axis representing the longitudinal catheter axis (see Fig 1). High intensity voltage pulses were applied in monopolar mode to the endocardium, as in González-Suárez *et al* [11].

### Governing equations

Laplace's Equation was used to compute the electrical potential $\phi$ [11]

$$\nabla(\sigma\nabla\phi) = 0 \tag{1}$$

where $\sigma$ is the electrical conductivity of the material. The electrical field distribution $E$ was calculated by $E = -\nabla\phi$, while $J$ the current density vector was calculated using the Ohm Law in its vector form:

$$J = \sigma E \tag{2}$$

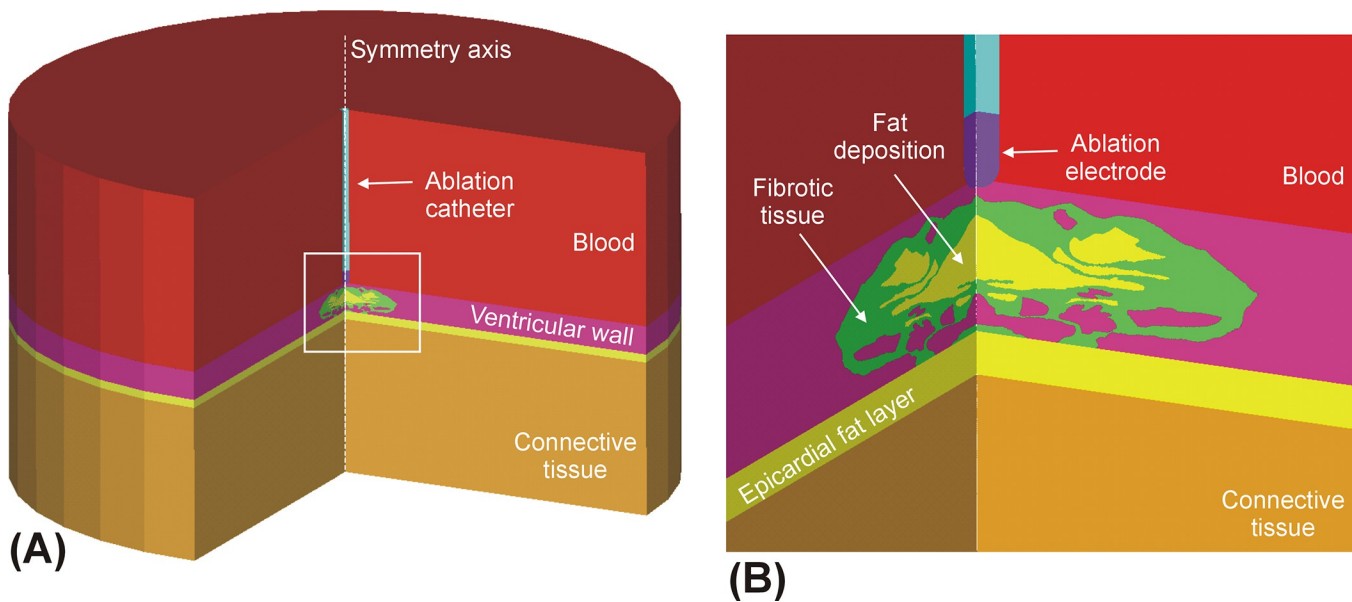

**Fig 1. A:** Model geometry in which the volume was created by rotating a 2D model around the catheter axis. **B:** Detail of the scar included in the ventricular wall (case of fat deposition in the scar).

We ignored the transient cellular response (membrane charging), as in previous cardiac PFA computer modeling studies [11–14]. This phenomenon takes a very short time and has no relevant effect on the PFA-induced lesion size. The model was numerically solved by the Finite Element Method on ANSYS software (ANSYS, Canonsburg, PA, USA). The ablation electrode voltage value was set to obtain delivered currents of 19, 22 and 25 A, while the outer boundaries were set to zero volts to mimic the monopolar mode. We set the current values as in the Centauri System (Galaxy Medical, CA, USA), a commercially available PFA generator that can work with focal cardiac ablation catheters such as the one modeled in this study. This meant that the applied voltage really varied in a range of ~200 V, according to the tissue's electrical characteristics. Instead of setting current, other PFA systems set the peak voltage value, so that the current depends on the tissue's electrical characteristics. The choice of setting current or voltage was irrelevant in the context of our study, which did not aim to predict the PFA-induced lesions created with a specific PFA generator, but to mimic the impact of intramyocardial fat on the electric field distribution.

## Modeling the spatial distribution of tissues involved

Six models were built to consider the variability of the spatial distributions of myocardium, fat and fibrosis (see Fig 2). The first two models were based on a homogeneous cardiac wall (i.e. without scar or intramyocardial fat): A) homogeneous healthy myocardium with an epicardial fat, B) homogeneous healthy myocardium without an epicardial fat layer (replaced by connective tissue). The four other models included scar tissue: C) the original model published in [10] without considering fat deposition, i.e. replacing fat by fibrotic tissue, D) as in C by including fat deposition, E) as in C but including a channel of healthy myocardium just under the ablation electrode, and F) a *blocked* model, in which the subendocardial myocardium was replaced by fibrotic tissue. Note that there is connective tissue under these layers, as shown in Fig 1. Although these distributions are specific examples of the possibly infinite number of distributions (and thus may appear somewhat arbitrary), they mimic clinically relevant situations

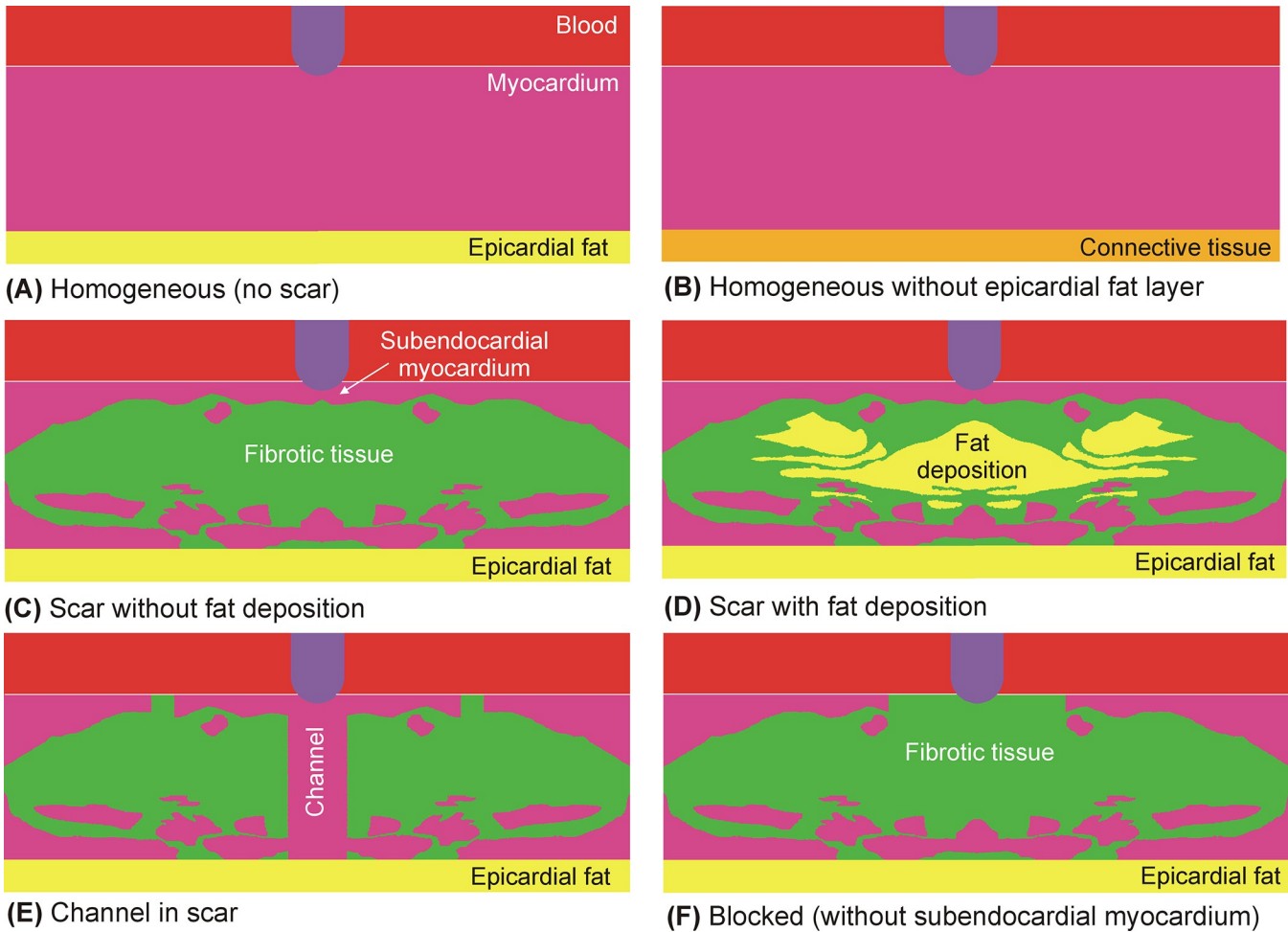

**Fig 2. Details of the modeled ventricular walls consisting of different tissues: Myocardium, fibrotic tissue and fat.**

in the context of scar-mediated VT ablation. The *channel* model was intended to mimic the scar de-channeling technique and consisted of ablating only the conduction channel entrances rather than an carrying out an extensive ablation [15]. The *blocked* model was intended to mimic a VT ablation with a deep arrhythmogenic focus and the fibrosis occupies almost all the space below the electrode (with very little subendocardial healthy myocardium). These six models together offer a representative sample for the analysis of the impact of fat on the electric field distribution. While models A and B are relevant in the context of PFA on the atrial wall to treat AF, models C−F are important in the context of PFA on substrates with intramyocardial fat, which could be related to a post-myocardial infarction scar.

### Tissue electrical properties

The change in σ induced by PFA was modeled by a sigmoid function [16] that depended on the electric field magnitude $\sigma(E)$ as follows:

$$\sigma(E) = \sigma_0 + \frac{\sigma_1 - \sigma_0}{1 + 10e^{-\frac{(|E|-58000)}{3000}}} \tag{3}$$

where $\sigma_0$ and $\sigma_1$ are the pre- and post-electroporation electric conductivities, respectively.

The values of $\sigma_0$ and $\sigma_1$ are thus related to the presence or absence of pores in the cell membrane. Although there are experimental studies on how electrical conductivity changes prior to and after PFA for tissues such as kidney [17] and liver [18], to our knowledge there are none for tissues involved in cardiac PFA, so that we had to make assumptions based on the current flow through the tissues and whether or not the cell membrane was electroporated. Prior to PFA, i.e. before the pores have been created, electric current flows only through the extracellular space as when the tissue is subjected to low frequency electrical excitation and the cell membrane acts as an electrical insulator. The $\sigma_0$ values are thus those measured at frequencies below beta dispersion (i.e. when the cell membrane impedes the current flow). In practical terms, $\sigma$ remains more or less constant between 1 and 10 kHz and increases at higher frequencies due to the electrical current flowing not only through the extracellular space but also through the cytoplasm. Following this reasoning, we chose the $\sigma_0$ and $\sigma_1$ tissue values as follows:

1. For healthy myocardium, we used the values reported in two *in vivo* experimental studies on a pig model [19, 20] in which $\sigma$ was measured for healthy and scar ventricular tissue in a broad range of frequencies (1–1000 kHz). These results showed that between 1 and 10 kHz, the $\sigma$ value was approximately 0.4±0.1 S/m, while at 1000 kHz, it rose to 0.6±0.1 S/m [19, 20]. This latter value is identical to that reported by Tsai *et al* [21] at 1 MHz, also measured in *in vivo* pig ventricle: 0.6±0.05 S/m. The two values 0.4 and 0.6 S/m were therefore used as the pre- ($\sigma_0$) and post-PFA ($\sigma_1$) electrical conductivity for healthy myocardium.

2. The scar (healed myocardium after infarction) contains very few surviving cells. In fact, the densely infarcted zone is composed almost exclusively of collagen [22]. This is reflected in the lack of a capacitive (i.e. frequency-dependent) response. In other words, there is no beta dispersion [19, 20, 22]. This means that for scar tissue, $\sigma_0$ and $\sigma_1$ can be considered to be identical, although there is experimental evidence suggesting that fibrotic scar is more conductive than healthy myocardium: [19, 20, 22, 23]. Salazar *et al* [20] and Cinca *et al* [19] reported values of around 0.8 and 0.9 S/m, respectively, while Schwartzman *et al* [22] obtained a value of 0.44 S/m at 40 kHz, as opposed to a lower value for the healthy tissue (0.14 S/m). Likewise, Fallert *et al* reported a value of 1.02 S/m at 15 kHz, as opposed to a lower value for healthy tissue (0.53 S/m) [23]. This tendency of fibrotic tissue to be more conductive has also been found in other organs such as rat liver, in which fibrous tissue (induced by a weekly intraperitoneal injection of dimethylnitrosamine in 45 male Sprague–Dawley rats) had an electrical conductivity $\geq$12% compared to that of normal control rats [24]. A value of 0.85 S/m was thus considered for fibrotic tissue, regardless of the electric field value.

3. We used the value of 0.08 S/m for fat reported by Gabriel *et al* [25], which was obtained from an *in vivo* pig model. Since this value was found to stay more or less constant within a broad frequency (10 kHz–1 MHz) [26], we assumed that $\sigma_0 = \sigma_1 = 0.08$ S/m. The tissue beyond the ventricular wall (which completed the limited domain model [11]), was assumed to be a mix of skeletal muscle and fat, as suggested by the CT images, with a ratio of fat varying from 20 to 80% [9]. Since this tissue is really far from the ablation electrode and hence considered to be unaffected by the electric field at any time, its electrical conductivity was that of the low frequency zone, i.e. lower beta dispersion. The electrical conductivity of skeletal muscle at low frequency was assumed to be 0.15 S/m [25]. The electrical conductivity of the tissue beyond the ventricular wall was thus $\sigma_0 = \sigma_1 = 0.115$ S/m (assuming a mix ratio of 50% of fat).

4. The blood circulation inside the ventricle cools the pool of blood cells around the ablation electrode. Even considering a broad range of possible blood velocities, e.g. from 0.1 to 100 cm/s, a blood cell subjected to the electric field around the ablation electrode will only have moved 0.1–100 μm during the application of a voltage pulse of 100 μs and less for shorter pulses. In other words, the blood cells surrounding the ablation electrode can also be assumed to be electroporated. However, we considered the same electrical conductivity of 0.6 S/m before and after PFA, since this value has been found to be more or less constant within a wide range of frequencies (100 Hz–1 MHz) [25].

Table 1 summarizes the data of electrical conductivity used in the model for the tissues studied.

## Assessment of the PFA-induced lesion size

An electric field value of 1000 V/cm was used as the irreversible electroporation lethal threshold, as in other previous PFA computer modeling studies [14, 27] and suggested by a recent experimental study based on a suspension *in vitro* model for cardiac cells [28]. We only plotted the PFA-induced lesion in the myocardium, since it is really the target of the PFA, so that no lethal threshold was considered for the other fibrotic and fat tissues.

## Results

Fig 3 shows the electrical field distributions for the six considered cases at a current of 22 A. There was very little difference in the computation time of the six models. Table 2 gives the PFA lesion size (depth and surface width) for the different ventricular wall models and three current values. Note that the lesion size was computed as the extension of the 1000 V/cm isoline, even though it was on tissue other than myocardium, i.e. occupying an area outside the ablation target. In general, the lesion size was similar for all the models except for the one with a fat deposition within the scar (Fig 3D), in which case the lesion was ~1 mm deeper than without this factor. As this was possibly due to the specific fat location and distribution, this result should not be valued quantitatively, but qualitatively. The most important finding is therefore the ability of the fat deposited in the scar to significantly alter the electric field distribution, something that is not found in fibrotic tissue. In fact, with fat present in the scar, the lesion depth did not change when the electric current was raised (Fig 4), and the deepest lesion limit (1000 V/cm isoline) always coincided with the lower fat boundary. The presence of fat also tends to widen the lesion, but this possibly depends on the specific shape of the deposit, since the lesion tends to follow the limits of the fat itself.

The alteration of the electric field caused by the presence of fat implied low electric field values at the current's entry and exit points in the fat zone (i.e. 'cold points' in terms of electric field), while high electric field values (i.e. electric field 'hot points') appeared in the fat's lateral areas (see Fig 5).

**Table 1. Electrical conductivities (S/m) of the tissues involved in the model pre- ($\sigma_0$) and post-electroporation ($\sigma_1$).**

| Tissue | $\sigma_0$ | $\sigma_1$ |
|---|---|---|
| Myocardium | 0.4 | 0.6 |
| Scar | 0.85 | 0.85 |
| Fat | 0.015 | 0.015 |
| Blood | 0.6 | 0.6 |

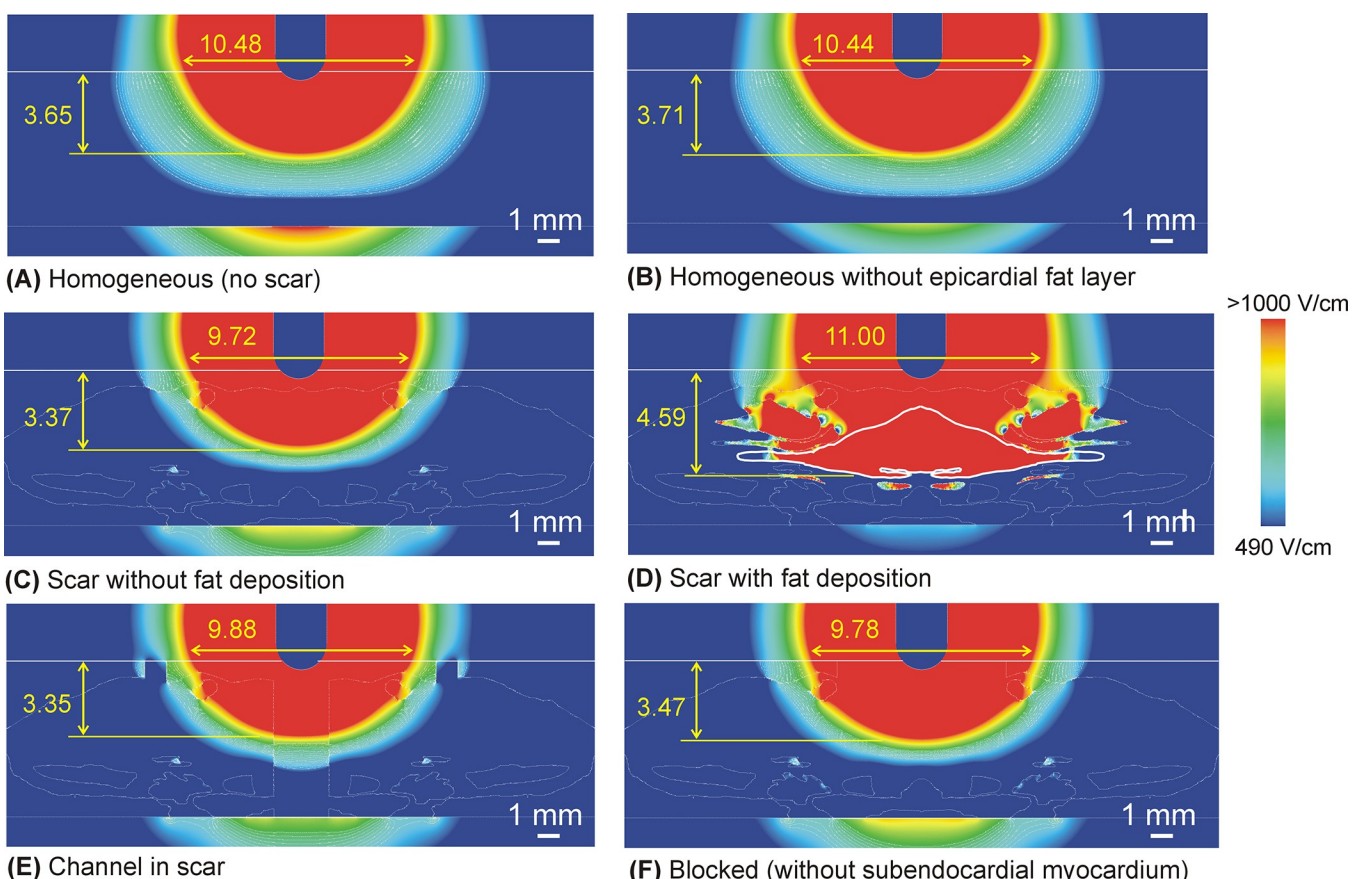

**Fig 3. Electric field distributions for the different cases considered (red color for >1000 V/cm and blue color for <490 V/cm) with 22 A current.** Lesion depth and surface width (in mm) induced by irreversible electroporation was computed with the 1000 V/cm isoline. In case D, the contour of the largest area of fat deposited in the scar is highlighted by a continuous thick white line.

The effect of the different ventricular wall models barely affected the voltage needed to deliver a given current. For example, the voltage varied between 1960 V (blocked model) to 2150 V (heterogeneous with fat deposition) at 22 A. The lesions became deeper and wider as the current was increased (except in the case of a scar with fat deposition), with an almost linear trend of 0.08 mm/A in depth and 0.25 mm/A in width.

**Table 2. Lesion sizes computed by 1000 V/cm isoline (D: Depth; SW: Surface width) for different delivered current and considered cases of ventricular wall.**

| | Applied current | | | | | |
|---|---|---|---|---|---|---|
| | 19 A | | 22 A | | 25 A | |
| Model | D (mm) | SW (mm) | D (mm) | SW (mm) | D (mm) | SW (mm) |
| **A) Homogeneous (no scar)** | 3.40 | 9.62 | 3.65 | 10.48 | 3.91 | 11.22 |
| **B) Homogeneous (no epicardial fat layer)** | 3.40 | 9.56 | 3.71 | 10.44 | 3.97 | 11.22 |
| **C) Scar without fat deposition** | 3.11 | 8.98 | 3.37 | 9.72 | 3.60 | 10.44 |
| **D) Scar with fat deposition** | 4.59 | 9.74 | 4.59 | 11.00 | 4.59 | 12.44 |
| **E) Channel in scar** | 3.11 | 9.08 | 3.35 | 9.88 | 3.58 | 10.60 |
| **F) Blocked** | 3.20 | 9.04 | 3.47 | 9.78 | 3.68 | 10.5 |

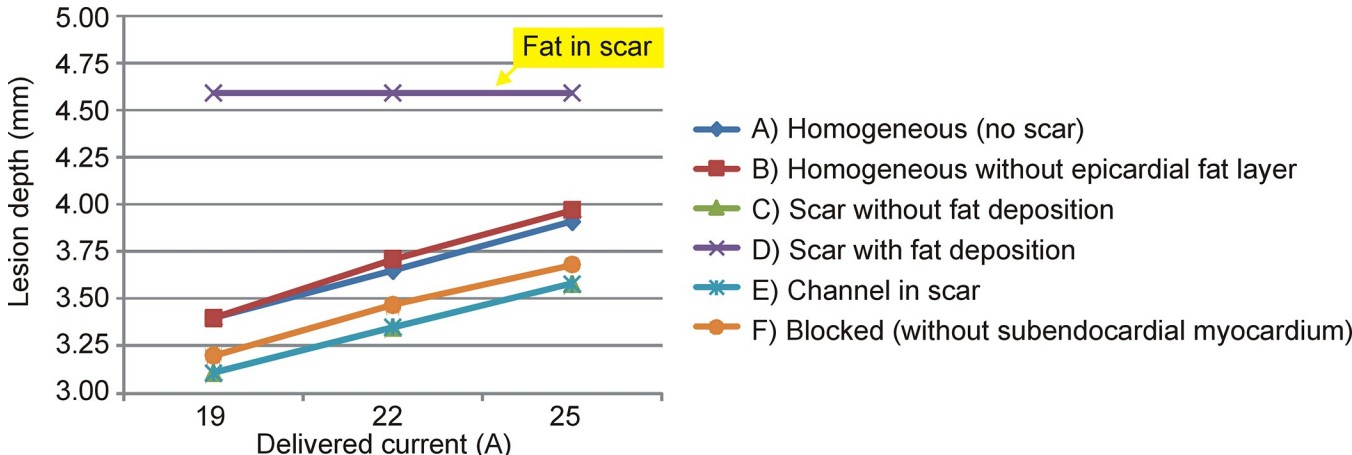

**Fig 4. Lesion depths induced by irreversible electroporation (computed with the 1000 V/cm isoline) for different delivered currents and for the six considered cases.** The lines showing the "channel in scar" and "scar without fat deposition" are almost identical.

## Discussion

The impact of fat on the electric field distribution and lesion shape in the myocardium (target) was analyzed by computer modeling. A set of the different spatial configurations of these tissues in the ventricular wall was considered to generate results that could be applied in clinical practice. Note that cases without a scar and intramyocardial fat can also be used to illustrate the impact of epicardial fat during PFA of atrial fibrillation.

### Qualitative description of the electrical impact of intramyocardial fat

We found that the presence of fibrotic tissue has little impact on the electric field distribution and lesion size compared to that of healthy myocardium only, which has already been suggested by experimental data [3]. However, we also found that the fat inside the scar significantly alters the electrical field distribution and the resulting lesion shape. To be more precise, the electric field tends to be higher in fat than in other tissues, even when the fat is far from the ablation electrode, so that 'cold points' (i.e. with a low electric field) appear around the fat electrical current entry and exit points, while 'hot points' (i.e. high electric field) appear in the lateral areas.

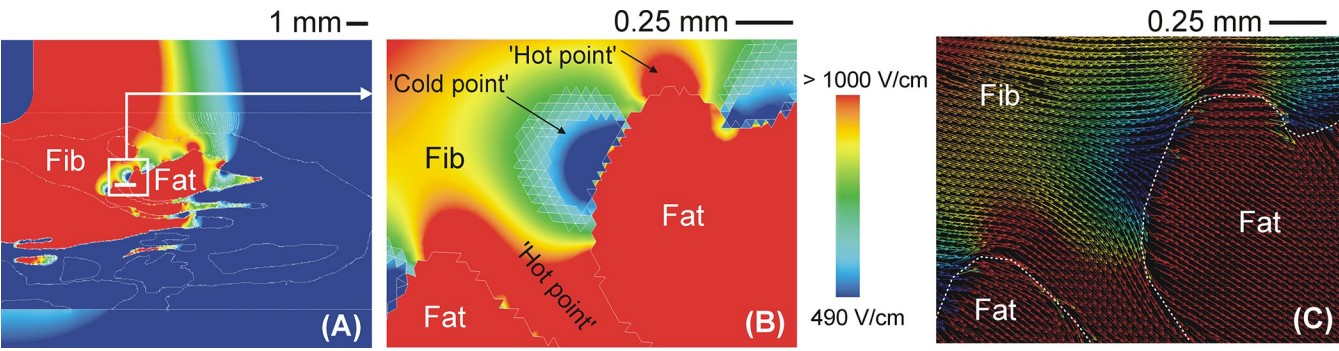

**Fig 5. A:** Electric field distribution for heterogeneous model with fat deposition (22 A current). Detail of electric field (**B**) and electric field vector (**C**) around fat zones, showing 'cold points' and 'hot points' since electrical current tends to bypass fat due to its lower electrical conductivity.

Our findings on the important role of intramyocardial fat during PFA are closely connected to the heterogeneity of the tissues in the ventricular scar in terms of electrical conductivity, with healthy myocardium and fibrosis offering similar values (0.6 and 0.85 S/m), while fat has a significantly lower value (0.08 S/m), which means the fat acts as an 'attractor' of the electric field, where it tends to be especially high. This 'attraction' seriously alters the electric field distribution in the contiguous tissues surrounding the fat.

This alteration is characterized by zones with a low electric field ('cold points') around the fat at the points where the electric current enters and exits and in the zones with a high electric field ('hot points') in the lateral areas (see Fig 5B). From the point of view of spatial distribution, this is due to the fact that the electric currents try to avoid the less conductive areas, such as those in the direction of the electric field vector in Fig 5C (current density vector has the same direction as the electric field when displacement currents are neglected, as shown in Eq (3)). The alteration of the electric field around structures surrounded by tissues with different electrical conductivity has already been described in the context of PFA. In essence, two circumstances can occur: 1) a structure with lower electrical conductivity than the surrounding tissue (as in our case with fat surrounded by fibrosis) or 2) a structure with higher electrical conductivity than the surrounding tissue (as occurs in blood vessels surrounded by fat [29] or by liver parenchyma [30]). In the former case, the 'cold points' appear at the current's entry and exit points (since it tries to avoid the less conductive structure), while in the latter case they appear at the sides, and 'hot points' occur at the current entry and exit points (since it tends to flow through the more conductive structure). This phenomenon has been computationally studied in the context of Low-Energy Defibrillation (LED), which is based on the 'Virtual Electrode' (VE) effect, in which the conductivity heterogeneities cause localized regions of depolarization and adjacent hyper-polarization [31]. This is caused by the electrical heterogeneity of the cardiac wall anatomy and the electrical boundary conditions to be applied between the interfaces with very different electrical conductivity values, as between fat and myocardium. In short, and as can be seen in Fig 6, by assuming that the displacement current is negligible compared to the conductive current, the current density $J$ and electric field $E$ vectors are simply related to each other by means of Ohm's Law in vector form, $J = \sigma \cdot E$. The two first electrical boundary conditions come from applying Maxwell's Equations at the interface: 1) the tangential component of $E$ ($E_t$) is continuous across the interface, i.e. the tangential value is the same in the two adjoining zones $E_{t,myo} = E_{t,fat}$; and 2) the normal (i.e. perpendicular)

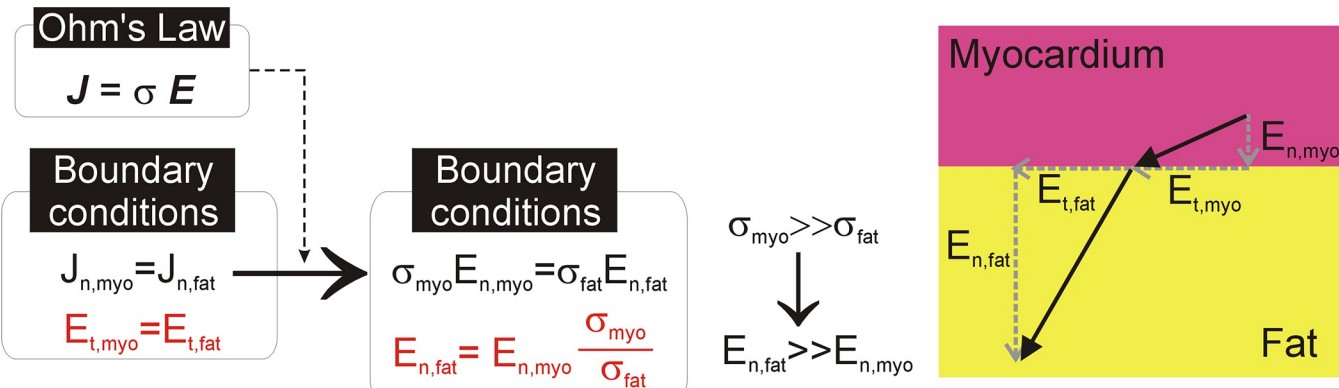

**Fig 6. Mathematical justification of the boundary conditions of normal and tangential components (in red) of the electric field vector $E$ at the interface between myocardium and fat.** As a result, the normal component of $E$ in the fat is much greater than the normal component in myocardium, which implies that the electrical field vector is much greater in fat.

component of $J$ ($J_n$) is the same in the two adjoining zones, i.e. $J_{t,myo} = J_{t,fat}$. By applying Ohm's Law to the second boundary condition, we find that the normal component of $E$ ($E_n$) is not continuous across the interface and is governed by the following equation: $\sigma_{myo} \cdot E_{n,myo} = \sigma_{fat} \cdot E_{n,fat}$, i.e by the ratio between the fat and myocardium electrical conductivities. Since $\sigma_{fat} << \sigma_{myo}$, $E_{n,fat} >> E_{n,myo}$, resulting in a higher electric field vector $E$ in fat.

In practical terms, we found that under certain circumstances such as those shown in Fig 2D, the fat could prevent damage to deeper tissues, maintaining a constant depth of 4.59 mm, despite current value increases from 19 to 25 A (see Fig 4). And not only deeper tissues, the mere presence of fat in the path of the electric current implies that contiguous tissues, even when they are closer to the surface, i.e. closer to the ablation electrode, could be subjected to a lower electric field. In other words, the electrical field tends to peak in fat zones. This can be clearly seen in all the cases in which the electric field was higher in the epicardial fat layer than in the contiguous myocardium, despite being deeper (see Fig 3A, 3C–3F). In the case without an epicardial fat layer we found the same phenomenon, since the connective tissue (see Fig 3B) was also considered to be less conductive than myocardium (0.11 vs. 0.6 S/m). The physical explanation for this is that the electric field is the gradient vector of the voltage, as mathematically expressed in Eq (2). The gradient vector represents the direction and magnitude of the spatial variation of a scalar magnitude, voltage in this case, so that despite the fact that the voltage (electric potential) from the electrode (at ~2 kV) to the dispersive electrode (at 0 V) gradually decreases, as could be expected, this decrease can be uneven according to the electrical conductivity of the tissues and is more abrupt in less conductive tissues such as fat (see Fig 7A). This more abrupt change results in a higher electric field value and is the reason why we can find points far from the ablation electrode with higher electric field values than other closer points (see Fig 7B).

## Comparison with experimental results and clinical implications

There are not yet enough published experimental data on PFA-induced lesion sizes with which to compare our computational findings. The preliminary results suggesting that PFA is able to ablate viable myocardium separated from the catheter by collagen and fat [5, 32] could be considered as not in line with our findings, since our results show that the fat concentrates high electric field values and prevents the lesion from reaching outlying tissues. However, our results do not show that the presence of fat always has a negative impact. In fact, the lesion was deeper and wider in the specific model considered with the presence of fat (as shown in Fig 3). What we have learned from our study is the way fat alters the surrounding electric field. The clinical implication in terms of PFA effectiveness will depend mainly on the spatial distribution of the fat and its relative position with respect to the ablation electrode and the ablation target. Note that the arrhythmogenic target could be in a 'hot or cold' spot in terms of the electric field, according to the above-mentioned factors, which do not seem to be easy to control in an experimental setup.

Also, there is currently a consensus that, since the different commercially available PFA generators work with very different undisclosed energy distribution protocols (electrode design, waveform and frequency, polarity, etc.), the experimental results obtained with a given setup are only valid for that specific case [33], making comparisons much more complicated. In this regard, the few published experimental studies used different technologies to that modeled by us: while we modeled a standard-tip catheter, Younis *et al* [5] conducted monopolar PFA with a relatively large spherical electrode (9-mm diameter) and Higuchi *et al* [32] conducted bipolar PFA with 4 band electrodes on an 8Fr catheter.

Apart from these differences, the worth of our study lies more in the qualitative description based on physical laws of how intramyocardial fat can alter the electric field distribution,

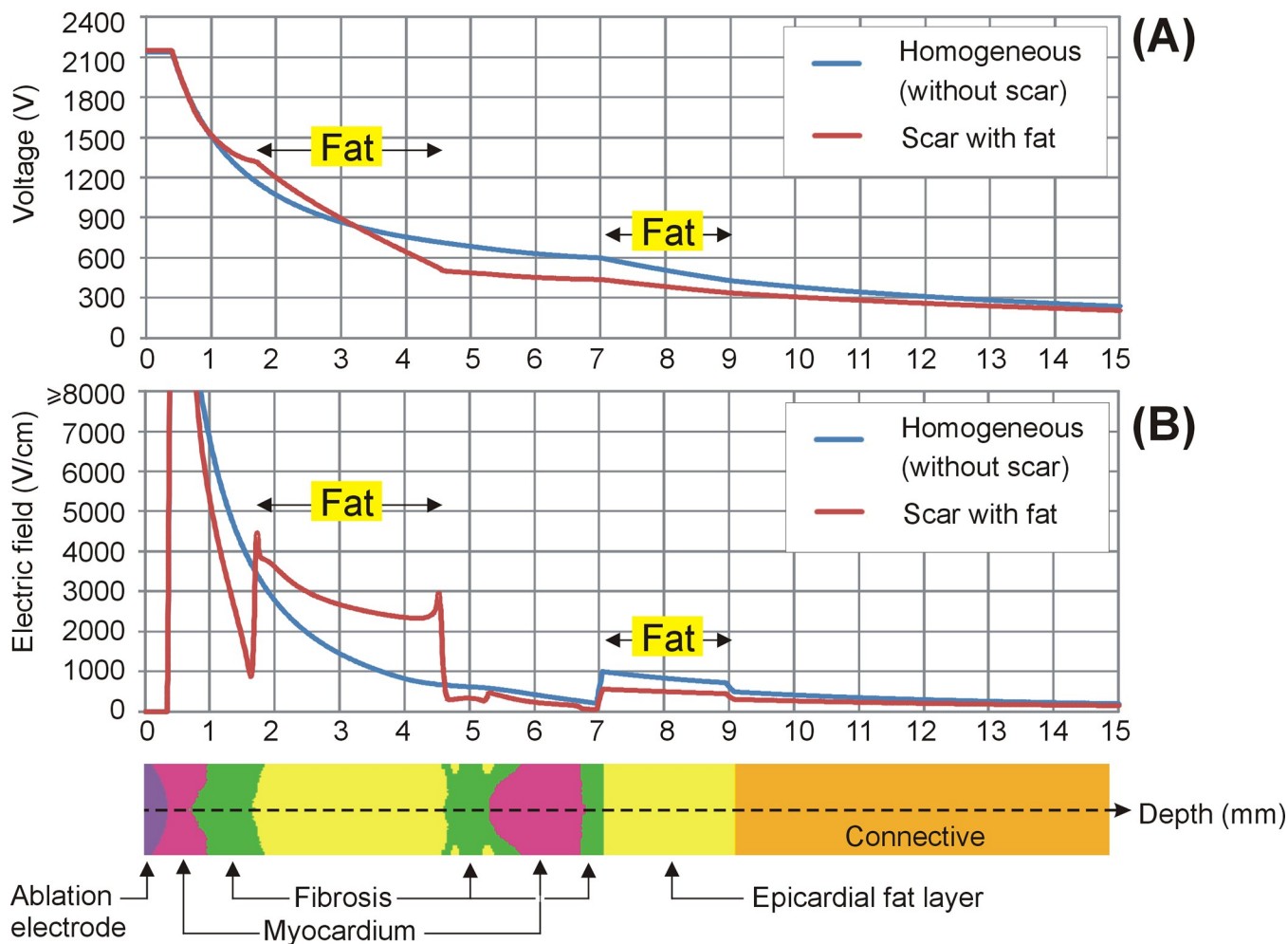

**Fig 7.** Variation of voltage (**A**) and electric field (**B**) along the axis under the electrode, from the tissue surface (0 mm, inside the electrode) to a depth of 15 mm, for two ventricular walls: homogeneous myocardium (i.e. without scar), and scar with fat deposition. Note that the voltage drop (i.e. how fast it falls along the axis) is greater across the less conductive tissue (fat), which results in electric field values even higher than those in tissues closer to the ablation electrode (myocardium and fibrosis).

regardless of the generator or protocol used. The concepts derived from our results on how fat alters the electric field should thus be valid, regardless of the PFA technical details, and our results do in fact suggest that some of these details related to the relative position of the fat and ablation electrode (such as electrode design and polarity) are possibly relevant to determine whether the presence of fat can favor or limit the lesion size.

Finally, since our results suggest that the position of 'hot and cold points' in terms of electric field depends mainly on the fat distribution, it is important to note that the distribution is highly dependent on the specific disease being treated. While fatty infiltration tends to be compact in the case of myocardial infarction, it is more disperse in arrhythmogenic right ventricular cardiomyopathy [34]. Our mathematical framework is based on valid physical laws (Laplace's Equation) regardless of the specific fat distribution considered. In this regard, although our model of infiltrated fat in the scar (Fig 2D) was initially inspired by a histological image derived from a post-infarction scar [9], it also includes specific zones in which the fat appears patchy, i.e. more disperse and separated by fibrotic tissue (see detail in Fig 5A). It is precisely around these small zones of scattered fat in which the electric field is significantly

altered, depending on the relative position of the ablation electrode, or what is the same, on the direction of the electric current, as shown in Fig 5B. In conclusion, what our results suggest is that when the fat is in the form of isolated areas such as those found in some non-ischemic cardiomyopathies, the alternation of 'hot and cold points' in the contiguous tissue in terms of electric field strength could compromise the efficacy of the PFA of arrhythmogenic foci around the fat.

## Limitations

This study only considered a specific ablation catheter with a 3.5-mm blunt tip, placed perpendicular to the tissue and inserted 0.5 mm, while only some specific spatial arrangements of the fat, fibrosis and myocardium were considered. This means that the sizes of the PFA-induced lesions predicted by the model cannot be directly compared to those from pre-clinical studies, especially due to the lethal threshold we used (1000 V/cm), possibly being dependent on the specific waveform, voltage setting and the number of applications. Despite this, we consider that this method is able to demonstrate the impact of intramyocardial fat on the electrical field distribution during PFA.

In the study we ignored the thermal side effects. Although they are theoretically negligible, since PFA is a nonthermal ablation technique, some heating can be expected to be induced by the electric field and current density. Even assuming that a temperature rise would also raise the electrical conductivity and somehow alter our results, the impact is probably minimum, since: 1) the fat's temperature coefficient is not very different to the rest of the tissues (1.6 −1.7%/˚C) [35], suggesting that the effect of heating on the electrical conductivity will be similar for all the tissues, and 2) the fat's tendency to show high electric field values, $E$ also implies low values of current density $J$, suggesting that Joule heating ($J·E$) is not expected to be different in fat to that of other tissues, as has been found in RFA [10].

We also ignored the anisotropy of the electrical conductivity. Although electrical anisotropy has been suggested to be important in fibrous muscles such as skeletal and cardiac, it only seems to be relevant at a low frequency, i.e. when the cell membrane plays an important role in the electrical terms, but not at a high frequency, when it is already virtually bypassed. In this regard, Tsai *et al* [21] observed that ventricular tissue at 10 kHz was slightly more conductive in the transverse than the longitudinal direction: 0.4 vs. 0.36 S/m, respectively. This implies a degree of anisotropy of only 11% in terms of electrical conductivity at low frequencies, which probably has a minimal impact on ablation modeling. Differences of around 26% were reported by Gabriel *et al* [25] at 40–70 Hz in three different directions, which implies an anisotropy ratio of 1.27, and much lower than those assumed by Zang *et al* [13] (between 1.43 and 6.25) in a computational PFA modeling study, who did find differences in terms of the lesion size induced by PFA between an isotropic and anisotropic model.

However, although anisotropy of the electrical conductivity has been suggested to be a relevant characteristic in the electroporation of breast tumors [36], there are still no experimental data to support a degree of anisotropy in cardiac tissue high enough to significantly affect the electric field distribution during PFA. It should be noted that this does not mean that anisotropy has no impact on the electric field distribution and PFA-induced lesion size, but rather that its effect is possibly not important in the context of our objective of showing how the presence of fat alters the electric field distribution.

Finally, we used a limited-domain model, i.e. only fragments of tissue around the ablation electrode were included, instead of a realistic reconstruction of the heart. This is a valid approach since the electric field values are practically negligible outside the limits considered, as shown in Fig 7.

## Conclusions

While the presence of fibrotic tissue has little impact on the electric field distribution and lesion size compared to that in healthy myocardium, intramyocardial fat significantly alters the electrical field distribution and the resulting lesion shape. In particular, the electric field tends to be higher in fat, even when the fat is farther from the ablation electrode. 'Cold points' with a low electric field appear around the fat at the electrical current's entry and exit points, while 'hot points' are formed in the lateral areas with a high electric field.

In practical terms, the positive or negative effect of fat on the PFA-induced lesion size will depend on the spatial distribution of the fat and its position relative to the electrode, making it impossible to draw a general conclusion on the presence of fat, while the same time suggesting the value of computational modeling as a predictive tool for planning PFA on substrates with high fat content.

## Supporting information

**S1 Data.**
(XLSX)

## Author Contributions

**Conceptualization:** Ana González-Suárez.

**Data curation:** Juan J. Pérez.

**Formal analysis:** Juan J. Pérez, Ana González-Suárez.

**Funding acquisition:** Juan J. Pérez, Ana González-Suárez.

**Methodology:** Juan J. Pérez, Ana González-Suárez.

**Writing – original draft:** Juan J. Pérez, Ana González-Suárez.

**Writing – review & editing:** Juan J. Pérez, Ana González-Suárez.

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
