## [Decision Letter · Decision Letter 0]

5 May 2023

PONE-D-23-07892How intramyocardial fat can alter the electric field distribution during Pulsed Field Ablation (PFA): Qualitative findings from computer modelingPLOS ONE

Dear Dr. González-Suárez,

Thank you for submitting your manuscript to PLOS ONE. After careful consideration, we feel that it has merit but does not fully meet PLOS ONE’s publication criteria as it currently stands. Therefore, we invite you to submit a revised version of the manuscript that addresses the points raised during the review process.

We look forward to receiving your revised manuscript.

Kind regards,

Kadiam Venkata Subbaiah, Ph.D

Academic Editor

PLOS ONE

“Funded by the Spanish Ministerio de Ciencia e Innovación / Agencia Estatal de Investigación (MCIN/AEI/10.13039/501100011033) with Grant number PID2022-136273OB-C31 (JJP) and PID2022-136273OA-C33 (AGS).”

Additional Editor Comments:

Dear Dr. Ana González-Suárez,

I am pleased to inform you that the above-referenced manuscript is of interest and potentially acceptable for publication, but a minor revision is required to meet the concerns of the reviewers. The reviewers' comments are provided below.

Reviewer-1:

In this original article, Pérez and González-Suárez aimed to evaluate the effect of electrical impact of the

fat deposited in the ventricular scar during PFA by means of computer modeling. Computer

models were built considering a PFA 3.5-mm blunt-tip catheter in contact with a 7-mm

ventricular wall (with and without a scar) and a 2-mm epicardial fat layer. The authors should be congratulated on conducting such an important study. The design is fascinating and very inclusive with 6 different scenarios. They have found that intramyocardial fat can alter the electric field distribution positively resulting in deeper lesion size during PFA when compared with tissue without fat. This may be explained by its much lower electrical conductivity than myocardium and fibrotic tissue.

The paper is well written, very relevant to the field of electrophysiology, and very novel. Here are some minor comments:

1. Abbreviations: words should be abbreviated on their first mention and then the abbreviation should be used throughout the entire manuscript. For example in line 6 you say radiofrequency ablation without abbreviation, and then at the end of the introduction (I wished the lines were numbered, would have been easier) you use it abbreviated. But then again in the methods you use full wording; radiofrequency ablation. Please review the manuscript and try to be consistent while using all abbreviations.

2. In your conclusion you state that fat can alter the size. Alter can mean a negative impact (smaller size) and positive impact (bigger lesions). I would revise it to highlight that fat alters the lesions size "positively" resulting in deeper and wider lesions.

3. Fat is not only in the ventricle! The discussion and introduction somehow focus on VT, but that's a small portion of our patients. The majority of our patients are atrial fibrillation patients. Several studies have demonstrated the presence of endocardial and epicardial fat depositions within the atria of patients with atrial fibrillation. I would comment on that and make the paper less oriented to VT only.

Reviwer-2:

This article explores the potential influence of intramyocardial fat on the effects of PFA ablation; the work is based entirely on in silico data. The key finding of this study is that intramyocardial fat alters the distribution of the electric field in multiple ways – including a “shielding” effect of deeper structures as well as the creation of various “hot” and “cold” spots that effectively receive more and less ablative energy. Also notable is the finding that the shielding effect may persist despite current dose escalation.

The authors could significantly boost the relevance of this work by expanding the repertoire of myocardial models in addition to model “D.” In particular, it would be interesting to see how PFA performs in the context of various non-ischemic cardiomyopathies, some of which exhibit particularly strong propensity toward fibrofatty replacement (e.g. ARVC). At the very least, it would be helpful to explore other scenarios where the fatty layer is not essentially compact (as seen in model “D”) but more scattered. Would the same findings be encountered?

It is also important to note that this manuscript’s findings are generally NOT in line with the currently published data. This includes Younis at al. 2022 (PMID 36194542) which the authors cite and Higuchi et al. 2022 (PMID 36779624). While that does not mean that the findings of this work are incorrect, the authors should try to find a way to reconcile this difference. The authors do state that variations in PFA technology make comparisons difficult (p16), the basic “fat shielding” concept should presumably hold regardless of the pulse waveform, frequency and polarity (monopolar vs bipolar) used.

Reviewer-3:

The present study is based on a computer model built from realistic images of histological samples of the cardiac wall, which include intact myocardial tissue, infiltrated fat, and fibrotic tissue derived from a previous myocardial infarction. Despite the limitations inherent to any computer modeling study, the authors present interesting results that are difficult to obtain through in vivo or clinical experimentation. Although the conclusions regarding the impact of fibrosis have already been previously suggested in other studies, the conclusions regarding the impact of fat are novel but also controversial, since there are clinical studies that suggest that PFA would be capable of creating lesions through intramyocardial fat. While this may be true in some circumstances, the fact is that clinical and experimental studies inherently lack control over the real spatial distribution of fat, making it impossible to draw generic or biophysically based conclusions. In contrast, computer modeling based on physical laws of bioelectricity shows that fat has an important impact on the distribution of the electric field. For this reason I consider that the results of this study are novel, relevant and with a clinical impact because they encourage that more experimental studies should be carried out in order to determine the real potential of PFA on cardiac substrates that contain fat.

I have some comments about the manuscript which should be considered:

1) At present, there are different equipments to carry out electropolation of biological tissue, and each manufacturer has its own waveform protocol with its specific characteristics. The authors could suggest what impact the different protocols would have on their results and conclusions. For example, the use of bipolar versus monopolar protocols, or short (2 microseconds) versus long (100 microseconds) pulse widths.

2) There is also a current controversy in relation to how electroporation pulses could affect the electrical conduction system of the heart. Could the authors expand their discussion to include the possible impact of PFA on these types of tissues and its relation with the fat presence?

3) The authors solved a static electrical problem by ignoring the current transient that is generated when the pulses are applied. I suggest including an additional comment about how this could affect the conclusions of the study.

4) Although the authors justify that the spatial distributions of tissues shown in Fig. 2 are sufficiently representative of the clinical cases that can be found in PFA ablation, I wonder if there could be any case in which the electrode is situated almost directly on adipose tissue, and what would be the impact of this in the light of the results of the modeling study.

Reviewers' comments:

Reviewer's Responses to Questions

**Comments to the Author**

1. Is the manuscript technically sound, and do the data support the conclusions?

Reviewer #1: Yes

Reviewer #2: Yes

Reviewer #3: Yes

2. Has the statistical analysis been performed appropriately and rigorously? 

Reviewer #1: N/A

Reviewer #2: N/A

Reviewer #3: N/A

3. Have the authors made all data underlying the findings in their manuscript fully available?

Reviewer #1: Yes

Reviewer #2: Yes

Reviewer #3: Yes

4. Is the manuscript presented in an intelligible fashion and written in standard English?

Reviewer #1: Yes

Reviewer #2: Yes

Reviewer #3: Yes

5. Review Comments to the Author

Reviewer #1: In this original article, Pérez and González-Suárez aimed to evaluate the effect of electrical impact of the

fat deposited in the ventricular scar during PFA by means of computer modeling. Computer

models were built considering a PFA 3.5-mm blunt-tip catheter in contact with a 7-mm

ventricular wall (with and without a scar) and a 2-mm epicardial fat layer. The authors should be congratulated on conducting such an important study. The design is fascinating and very inclusive with 6 different scenarios. They have found that intramyocardial fat can alter the electric field distribution positively resulting in deeper lesion size during PFA when compared with tissue without fat. This may be explained by its much lower electrical conductivity than myocardium and fibrotic tissue.

The paper is well written, very relevant to the field of electrophysiology, and very novel. Here are some minor comments:

1. Abbreviations: words should be abbreviated on their first mention and then the abbreviation should be used throughout the entire manuscript. For example in line 6 you say radiofrequency ablation without abbreviation, and then at the end of the introduction (I wished the lines were numbered, would have been easier) you use it abbreviated. But then again in the methods you use full wording; radiofrequency ablation. Please review the manuscript and try to be consistent while using all abbreviations.

2. In your conclusion you state that fat can alter the size. Alter can mean a negative impact (smaller size) and positive impact (bigger lesions). I would revise it to highlight that fat alters the lesions size "positively" resulting in deeper and wider lesions.

3. Fat is not only in the ventricle! The discussion and introduction somehow focus on VT, but that's a small portion of our patients. The majority of our patients are atrial fibrillation patients. Several studies have demonstrated the presence of endocardial and epicardial fat depositions within the atria of patients with atrial fibrillation. I would comment on that and make the paper less oriented to VT only.

4.

Reviewer #2: This article explores the potential influence of intramyocardial fat on the effects of PFA ablation; the work is based entirely on in silico data. The key finding of this study is that intramyocardial fat alters the distribution of the electric field in multiple ways – including a “shielding” effect of deeper structures as well as the creation of various “hot” and “cold” spots that effectively receive more and less ablative energy. Also notable is the finding that the shielding effect may persist despite current dose escalation.

The authors could significantly boost the relevance of this work by expanding the repertoire of myocardial models in addition to model “D.” In particular, it would be interesting to see how PFA performs in the context of various non-ischemic cardiomyopathies, some of which exhibit particularly strong propensity toward fibrofatty replacement (e.g. ARVC). At the very least, it would be helpful to explore other scenarios where the fatty layer is not essentially compact (as seen in model “D”) but more scattered. Would the same findings be encountered?

It is also important to note that this manuscript’s findings are generally NOT in line with the currently published data. This includes Younis at al. 2022 (PMID 36194542) which the authors cite and Higuchi et al. 2022 (PMID 36779624). While that does not mean that the findings of this work are incorrect, the authors should try to find a way to reconcile this difference. The authors do state that variations in PFA technology make comparisons difficult (p16), the basic “fat shielding” concept should presumably hold regardless of the pulse waveform, frequency and polarity (monopolar vs bipolar) used.

Reviewer #3: The present study is based on a computer model built from realistic images of histological samples of the cardiac wall, which include intact myocardial tissue, infiltrated fat, and fibrotic tissue derived from a previous myocardial infarction. Despite the limitations inherent to any computer modeling study, the authors present interesting results that are difficult to obtain through in vivo or clinical experimentation. Although the conclusions regarding the impact of fibrosis have already been previously suggested in other studies, the conclusions regarding the impact of fat are novel but also controversial, since there are clinical studies that suggest that PFA would be capable of creating lesions through intramyocardial fat. While this may be true in some circumstances, the fact is that clinical and experimental studies inherently lack control over the real spatial distribution of fat, making it impossible to draw generic or biophysically based conclusions. In contrast, computer modeling based on physical laws of bioelectricity shows that fat has an important impact on the distribution of the electric field. For this reason I consider that the results of this study are novel, relevant and with a clinical impact because they encourage that more experimental studies should be carried out in order to determine the real potential of PFA on cardiac substrates that contain fat.

I have some comments about the manuscript which should be considered:

1) At present, there are different equipments to carry out electropolation of biological tissue, and each manufacturer has its own waveform protocol with its specific characteristics. The authors could suggest what impact the different protocols would have on their results and conclusions. For example, the use of bipolar versus monopolar protocols, or short (2 microseconds) versus long (100 microseconds) pulse widths.

2) There is also a current controversy in relation to how electroporation pulses could affect the electrical conduction system of the heart. Could the authors expand their discussion to include the possible impact of PFA on these types of tissues and its relation with the fat presence?

3) The authors solved a static electrical problem by ignoring the current transient that is generated when the pulses are applied. I suggest including an additional comment about how this could affect the conclusions of the study.

4) Although the authors justify that the spatial distributions of tissues shown in Fig. 2 are sufficiently representative of the clinical cases that can be found in PFA ablation, I wonder if there could be any case in which the electrode is situated almost directly on adipose tissue, and what would be the impact of this in the light of the results of the modeling study.

6. PLOS authors have the option to publish the peer review history of their article (what does this mean?). If published, this will include your full peer review and any attached files.

Reviewer #1: **Yes: **Arwa Younis

Reviewer #2: No

Reviewer #3: No

---

## [Author Response · Author response to Decision Letter 0]

2 Jun 2023

RESPONSE TO REVIEWERS

Manuscript title: How intramyocardial fat can alter the electric field distribution during Pulsed Field Ablation (PFA): Qualitative findings from computer modeling

Authors: Juan J. Pérez, Ana González-Suárez

ID manuscript: PONE-D-23-07892

The following are the individual responses to the reviewers’ comments and explain how the revised version of the manuscript has been changed accordingly. The new text is in red in the revised version.

REVIEWER #1

COMMENT #1: “Abbreviations: words should be abbreviated on their first mention and then the abbreviation should be used throughout the entire manuscript. For example in line 6 you say radiofrequency ablation without abbreviation, and then at the end of the introduction (I wished the lines were numbered, would have been easier) you use it abbreviated. But then again in the methods you use full wording; radiofrequency ablation. Please review the manuscript and try to be consistent while using all abbreviations.”

Change in manuscript: Done.

COMMENT #2: “In your conclusion you state that fat can alter the size. Alter can mean a negative impact (smaller size) and positive impact (bigger lesions). I would revise it to highlight that fat alters the lesions size "positively" resulting in deeper and wider lesions.”

Response: In the light of our results, we do not think that the presence of fat always has a positive impact, but that the effect will depend on the spatial distribution of fat in each case and its relative position with respect to the ablation electrode. As pointed out in the Discussion, under some circumstances the fat could prevent damage to deeper tissues, maintaining a constant depth even as the current increases. As we have emphasized, fat constitutes an attractor for high electric field values, creating consequently low values in adjacent tissues (myocardium or fibrosis). This implies that the electric field might be considerably reduced in the myocardium located between the ablation electrode and a relatively deep area of fat, which would have a negative impact in terms of a smaller lesion depth than the absence of fat (e.g. compare the red and blue lines in Fig. 7B at a depth of 1.5 mm).The opposite can also occur, as suggested by Fig. 3, where the lesion is greater in the presence of fat.

Change in manuscript: We understand the reviewer's concern about sending a message in terms of clinical applicability. However, our conclusions were established mainly in terms of the physical description of the phenomenon and not so much of the advantage or disadvantage of the presence of fat. We have added a sentence indicating that the variation in lesion size due to the presence of fat will itself depend on the spatial distribution of the fat and its position relative to the electrode (see subsection “Comparison with experimental results and clinical implications”).

COMMENT #3: “Fat is not only in the ventricle! The discussion and introduction somehow focus on VT, but that's a small portion of our patients. The majority of our patients are atrial fibrillation patients. Several studies have demonstrated the presence of endocardial and epicardial fat depositions within the atria of patients with atrial fibrillation. I would comment on that and make the paper less oriented to VT only.”

Change in manuscript: We have reoriented the Introduction, Methods and Discussion towards the general effect of fat during PFA, by considering both fat in the epicardium (with important implications for ablation of atrial fibrillation) and intramyocardial fat, i.e. deposited in the subendocardium zone, caused by a post-myocardial infarction scar or by other causes, such as cardiomyopathy [1].

References

1. Wang B, Wan X, Li Y, Xie M. Endocardial fibrosis complicated with a great amount of fat accumulation in subendocardium: a rare form of restrictive cardiomyopathy. Eur Heart J. 2019 Jun 1;40(21):1740-1741. doi: 10.1093/eurheartj/ehz162. PMID: 30907414.

REVIEWER #2

COMMENT #1: “The authors could significantly boost the relevance of this work by expanding the repertoire of myocardial models in addition to model “D.” In particular, it would be interesting to see how PFA performs in the context of various non-ischemic cardiomyopathies, some of which exhibit particularly strong propensity toward fibrofatty replacement (e.g. ARVC). At the very least, it would be helpful to explore other scenarios where the fatty layer is not essentially compact (as seen in model “D”) but more scattered. Would the same findings be encountered?”

Response: This is an extremely interesting point since not only the presence of infiltrated fat in the myocardium is considered arrhythmogenic [1], and hence a potential target for PFA, but also the spatial distribution of infiltrated fat is highly dependent on the specific disease, as you mention. While fatty infiltration tends to be compact in myocardial infarction, it appears more dispersed in the case of arrhythmogenic right ventricular cardiomyopathy [1]. Note that our mathematical framework is based on physical laws (Laplace’s equation, in this case) which are valid regardless of the specific fat distribution considered. In this regard, note that although our model “D” was initially inspired by a histological image derived from a post-infarction scar [2], it also includes specific zones in which the fat appears patchy, i.e. more dispersed and separated by fibrotic tissue (see detail in Fig. 5A). It does not seem to be necessary to modify/expand the model “D” to mimic scattered fat. 

Interestingly, it is precisely around these small zones of scattered fat where the electric field is significantly altered depending on the relative position of the ablation electrode, or what is the same, on the direction of the electric current, as shown in Fig. 5B. In conclusion, what our results suggest is that when the fat is in the form of isolated areas (see detail in Fig. 5A), such as those found in some non-ischemic cardiomyopathies, the alternation of ‘hot and cold points’ in the contiguous tissue in terms of electric field strength could compromise the efficacy of PFA of arrhythmogenic foci around the fat.

Change in manuscript: This comment has been added in the Discussion.

COMMENT #2: “It is also important to note that this manuscript’s findings are generally NOT in line with the currently published data. This includes Younis at al. 2022 (PMID 36194542) which the authors cite and Higuchi et al. 2022 (PMID 36779624). While that does not mean that the findings of this work are incorrect, the authors should try to find a way to reconcile this difference.”

Response: As mentioned, in addition to the few experimental results available, there is a discrepancy between the technologies used. For instance, while we modeled a standard-tip catheter, Younis et al [3] conducted monopolar PFA using a relatively large spherical electrode (Sphere-9, Affera, Inc), and Higuchi et al [4] conducted bipolar PFA using 4 band electrodes placed on an 8Fr catheter (Farapulse). Although these studies are by no means conclusive, they suggest that the effect of PFA extends from the subendocardium through collagen and fat to the epicardial layers, at least more effectively than RFA. 

Despite the fact that our results suggest that the presence of fat alters the electric field value in its surroundings (acting as an attractor for a high electric field value), its effect does not have to be understood as a limitation in terms of lesion size: Although it is true that in some circumstances the presence of fat prevents the lesion from increasing in size as the administered electrical current increases (Fig. 4), this does not prevent the depth of the lesion from being deeper, even with fat than without fat (see Fig. 3), which is in line with the few experimental studies published to date. In qualitative terms, the presence of fat in deep areas ‘tends to enlarge’ the lesion size (Fig. 7).

Change in manuscript: This comment has been added to the Discussion to explain the differences between our results and the published data.

COMMENT #3: “The authors do state that variations in PFA technology make comparisons difficult (p16), the basic “fat shielding” concept should presumably hold regardless of the pulse waveform, frequency and polarity (monopolar vs bipolar) used.”

Response: Completely agree. The concepts derived from the computational results on how fat alters the electric field should be valid regardless of the technical details of PFA (electrode design, waveform, timing, polarity, etc.). However, our results do suggest that some of these details (such as electrode design and polarity) could be relevant in determining whether a specific fat distribution will favor or limit the lesion size and therefore whether PFA will achieve the target.

Change in manuscript: The comment on this point has been expanded with this information.

References

1. Anumonwo JMB, Herron T. Fatty Infiltration of the Myocardium and Arrhythmogenesis: Potential Cellular and Molecular Mechanisms. Front Physiol. 2018 Jan 22;9:2. doi: 10.3389/fphys.2018.00002. PMID: 29403390; PMCID: PMC5786512.

2. Sasaki T, Calkins H, Miller CF, Zviman MM, Zipunnikov V, Arai T, Sawabe M, Terashima M, Marine JE, Berger RD, Nazarian S, Zimmerman SL: New insight into scar-related ventricular tachycardia circuits in ischemic cardiomyopathy: Fat deposition after myocardial infarction on computed tomography—A pilot study. Heart Rhythm 2015;12:1508-1518.

3. Younis A, Zilberman I, Krywanczyk A, Higuchi K, Yavin HD, Sroubek J, Anter E. Effect of Pulsed-Field and Radiofrequency Ablation on Heterogeneous Ventricular Scar in a Swine Model of Healed Myocardial Infarction. Circ Arrhythm Electrophysiol. 2022 Oct;15(10):e011209. doi: 10.1161/CIRCEP.122.011209. Epub 2022 Oct 4. PMID: 36194542.

4. Higuchi S, Im SI, Stillson C, Buck ED, Jerrell S, Schneider CW, Speltz M, Gerstenfeld EP. Effect of Epicardial Pulsed Field Ablation Directly on Coronary Arteries. JACC Clin Electrophysiol. 2022 Dec;8(12):1486-1496. doi: 10.1016/j.jacep.2022.09.003. Epub 2022 Oct 26. PMID: 36779624.

REVIEWER #3

COMMENT #1: “At present, there are different equipments to carry out electropolation of biological tissue, and each manufacturer has its own waveform protocol with its specific characteristics. The authors could suggest what impact the different protocols would have on their results and conclusions. For example, the use of bipolar versus monopolar protocols, or short (2 microseconds) versus long (100 microseconds) pulse widths.”

Response: As mentioned by the reviewer, each PFA generator has different technical specifications. The differences in protocol in terms of peak voltage, pulse duration and amount, timing of the bursts, polarity, etc. will impact the lethal threshold of electric field strength. Overall, the higher the energy (accumulated duration x peak voltage) the lower the lethal threshold. Moreover, the frequency (related to pulse duration) will affect the risk of electrical stimulation and thermal side-effects. Despite this, our conclusions on the impact of fat and the appearance of ‘hot and cold points’ will remain unchanged.

Change in manuscript: A comment on this point has been added to the Discussion (see also Comment #3 of Reviewer #2).

COMMENT #2: “There is also a current controversy in relation to how electroporation pulses could affect the electrical conduction system of the heart. Could the authors expand their discussion to include the possible impact of PFA on these types of tissues and its relation with the fat presence?”

Response: Although this is certainly an interesting issue, it is outside the scope of our current study. Note that our computer model considers only the differences in electrical conductivity between the tissues involved and not their vulnerability in terms of being electroporated by a given protocol. In this regard, future computer modeling studies could be planned by considering the presence of the cells responsible for the electrical conduction system in the same way we considered fat deposition in our model. Unfortunately, information on the electrical characterization (passive electrical properties) of these tissues is still very scarce.

COMMENT #3: “The authors solved a static electrical problem by ignoring the current transient that is generated when the pulses are applied. I suggest including an additional comment about how this could affect the conclusions of the study.”

Change in manuscript: During PFA there is an extra current demand involved in charging and discharging the cell membranes. If the PFA generator is properly designed, it must be capable of managing the extra current demanded without altering the value of the voltage during the pulse. In our study we assumed that the generator is sized to distribute constant current. These issues are highly dependent on the technical characteristics of individual generators and manufacturers, and in any case do not affect the conclusions of the study.

COMMENT #4: “Although the authors justify that the spatial distributions of tissues shown in Fig. 2 are sufficiently representative of the clinical cases that can be found in PFA ablation, I wonder if there could be any case in which the electrode is situated almost directly on adipose tissue, and what would be the impact of this in the light of the results of the modeling study.”

Response: This is an interesting point. As intramyocardial fat is usually infiltrated in subendocardial areas it does not seem reasonable to assume a direct fat/electrode contact in an endocardial approach. However, it is possible to find this condition during epicardial PFA. This approach is less usual than endovascular and is conducted with the minimal surgical incisions [1]. In a previous modeling study we already reported that the epicardial fat layer notably ‘trapped’ the electrical field around the electrode, drastically reducing the values beyond this layer, i.e. in the myocardial tissue.

References

1. González-Suárez A, O'Brien B, O'Halloran M, Elahi A. Pulsed Electric Field Ablation of Epicardial Autonomic Ganglia: Computer Analysis of Monopolar Electric Field across the Tissues Involved. Bioengineering (Basel). 2022 Nov 27;9(12):731. doi: 10.3390/bioengineering9120731. PMID: 36550937; PMCID: PMC9774172.

---

## [Editor Report · Decision Letter 1]

8 Jun 2023

How intramyocardial fat can alter the electric field distribution during Pulsed Field Ablation (PFA): Qualitative findings from computer modeling

PONE-D-23-07892R1

Dear Dr.Ana González-Suárez,

We’re pleased to inform you that your manuscript has been judged scientifically suitable for publication and will be formally accepted for publication once it meets all outstanding technical requirements.

Kind regards,

Kadiam Venkata Subbaiah, Ph.D

Academic Editor

PLOS ONE
---

## [Editor Report · Acceptance letter]

14 Jun 2023

PONE-D-23-07892R1 

How intramyocardial fat can alter the electric field distribution during Pulsed Field Ablation (PFA): Qualitative findings from computer modeling 

Dear Dr. González-Suárez:

I'm pleased to inform you that your manuscript has been deemed suitable for publication in PLOS ONE. Congratulations! Your manuscript is now with our production department. 

Kind regards, 

on behalf of

Dr. Kadiam Venkata Subbaiah 

Academic Editor

PLOS ONE